# `BoxLM`: Unifying Structures and Semantics of Medical Concepts for Diagnosis Prediction in Healthcare

Yanchao Tan [1 2 3]  Hang Lv [1 2 3]  Yunfei Zhan [1 2 3]  Guofang Ma [4 5]  Bo Xiong [6]  Carl Yang [7]

## Abstract

Language Models (LMs) have advanced diagnosis prediction by leveraging the semantic understanding of medical concepts in Electronic Health Records (EHRs). Despite these advancements, existing LM-based methods often fail to capture the structures of medical concepts (e.g., hierarchy structure from domain knowledge). In this paper, we propose `BoxLM`, a novel framework that unifies the structures and semantics of medical concepts for diagnosis prediction. Specifically, we propose a structure-semantic fusion mechanism via box embeddings, which integrates both ontology-driven and EHR-driven hierarchical structures with LM-based semantic embeddings, enabling interpretable medical concept representations. Furthermore, in the box-aware diagnosis prediction module, an evolve-and-memorize patient box learning mechanism is proposed to model the temporal dynamics of patient visits, and a volume-based similarity measurement is proposed to enable accurate diagnosis prediction. Extensive experiments demonstrate that `BoxLM` consistently outperforms state-of-the-art baselines, especially achieving strong performance in few-shot learning scenarios, showcasing its practical utility in real-world clinical settings.

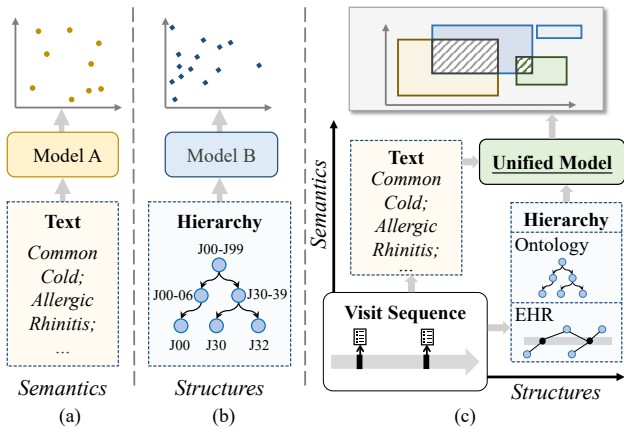

*Figure 1.* Exploring semantics and structures in healthcare. (a) modeling semantical similarity with vector embeddings. (b) encoding structural relationships with vector embeddings. (c) integrating semantic and structural dimensions with box embeddings.

## 1. Introduction

Accurate diagnosis prediction based on Electronic Health Records (EHRs) has emerged as a pivotal task in modern healthcare, enabling personalized treatment planning and improved patient outcomes. However, the widespread application of EHR-based systems is hindered by concerns over data privacy and limited accessibility to patient-specific datasets, resulting in the few-shot learning problem (Xu et al., 2024). This problem necessitates innovative approaches with limited patient EHR data while maintaining predictive performance. Recent advancements in Language Models (LMs) have demonstrated their potential to address the few-shot learning problem by incorporating the semantic understanding of medical concepts. LMs, such as BioBERT (Lee et al., 2020) and PubmedBERT (Gu et al., 2021), introduce pre-trained biomedical embeddings. Moreover, cutting-edge models like MedBench (Cai et al., 2024) and BioMistral (Labrak et al., 2024) further refine this approach through large-scale training on medical corpora, achieving notable improvements in semantic representation and contextual understanding of limited medical concepts.

Despite these advancements, existing LM-based methods often fail to capture the structures of medical concepts, such

---

[1]Engineering Research Center of Big Data Intelligence, Ministry of Education, Fuzhou, China [2]Fujian Key Laboratory of Network Computing and Intelligent Information Processing, Fuzhou University, Fuzhou, China [3]College of Computer and Data Science, Fuzhou University, Fuzhou, China [4]School of Computer Science, Zhejiang Gongshang University, Zhejiang, China [5]Zhejiang Key Laboratory of Big Data and Future E-Commerce Technology, Zhejiang, China [6]Center for Biomedical Informatics Research, Stanford University, Palo Alto, USA [7]Department of Computer Science, Emory University, Atlanta, USA. Correspondence to: Guofang Ma <maguofang@zjgsu.edu.cn>.

*Proceedings of the 42^nd International Conference on Machine Learning*, Vancouver, Canada. PMLR 267, 2025. Copyright 2025 by the author(s).

as the hierarchy of medical ontologies and the hierarchy of EHR-driven visit patterns. These structures are crucial for accurate diagnosis prediction (Luo et al., 2020; Lv et al., 2024). For instance, consider a patient with a history of *common cold* and *chronic sinusitis*. Based solely on semantics, LM-based models might predict the next diagnosis as *common cold* due to symptom similarity and its higher frequency in the training data. By unifying semantics with structures, the model can leverage hierarchical relationships to improve accuracy. In the ontology hierarchy, *allergic rhinitis* and *chronic sinusitis* are both classified under *other diseases of the upper respiratory tract*. Moreover, *chronic sinusitis* is often a downstream complication of poorly managed *allergic rhinitis* in the EHR hierarchy. Recognizing these relationships, the model can more accurately predict *allergic rhinitis* as the next diagnosis, illustrating the importance of incorporating structural knowledge.

However, effectively modeling these complex structures within the framework of LM-based diagnosis prediction remains a significant challenge. Standard embedding techniques, as shown in Figure 1(a) and Figure 1(b), represent concepts as single points in a vector space. These vector embeddings, while effective at capturing similarity relationships, fail to encode complex relationships such as inclusion relations inherent in the hierarchy. For instance, vector embeddings can model the similarity between *hypertension* and *cardiovascular diseases* by encoding semantic meaning. Whereas, they do not explicitly capture the hierarchical relationship where *cardiovascular diseases* encompasses *hypertension*. Inspired by the success of box embeddings across various domains (Vilnis et al., 2018; Huang et al., 2023; Lv et al., 2024; Lin et al., 2024), we find that they offer a promising solution by representing entities as high-dimensional hyperrectangles, where two boxes can clearly "enclose" or "intersect" each other. As shown in Figure 1(c), box embeddings are well-suited to capturing complex relationships, including hierarchical inclusion and set-based intersections, making them an ideal choice for unifying structures and semantics in medical concepts.

In this paper, we propose BoxLM, a novel Box-aware Language Model for diagnosis prediction. BoxLM leverages the expressive power of box embeddings to seamlessly unify structural and semantic representations of medical concepts, enabling precise diagnosis prediction. Specifically, our framework consists of two key modules: (1) a *structure-semantic fusion* module that integrates both ontology-driven and EHR-driven hierarchical structures with semantic embeddings from Pre-trained Language Models (PLMs), and (2) a *box-aware diagnosis prediction* module that quantifies the similarity between patient visits and Clinical Classifications Software (CCS) codes using the intersection volume of their boxes. By modeling medical concepts as high-dimensional hyperrectangles, BoxLM significantly enhances

prediction accuracy, outperforming state-of-the-art models.

In summary, we make the following contributions:

- *Unified BoxLM Framework.* We propose a novel framework that unifies structures and semantics of medical concepts via box embeddings, capturing the complex relationships among diagnoses, CCS codes, and patient visits at the conceptual level.

- *Effective Model Designs.* We introduce a structure-semantic fusion mechanism to jointly integrate ontology-driven and EHR-driven hierarchical structures with PLM-based semantic embeddings for medical concept representation. Furthermore, we propose an evolve-and-memorize patient box learning mechanism and a volume-based diagnosis prediction, to quantify complex relationships between the patient and CCS codes.

- *Extensive Experiments on Real EHR Datasets.* We conduct extensive experiments on real-world EHR datasets, demonstrating that BoxLM outperforms state-of-the-art models in both visit-level and code-level diagnosis prediction. Particularly, BoxLM achieves strong performance in few-shot learning scenarios, highlighting its practical applicability in real-world clinical settings.

## 2. The BoxLM Framework

### 2.1. Problem Definition and BoxLM Overview

Given a patient's EHR data, which consists of a sequence of visits, each associated with a set of medical concepts (e.g., diagnoses and CCS codes), the goal is to predict the relevant CCS codes for the patient's next visit. Notably, CCS is a classification system that groups diagnoses into clinically meaningful categories for analytical purposes, whereas a diagnosis refers to the identification of a specific disease or condition in a patient's visit.

Let a patient $p$'s EHR data be represented as a sequence of visits $\{v_1, v_2, \ldots, v_t\}$, where each visit $v_t$ is associated with a set of medical concepts, including diagnoses $d \in \mathcal{M}_d$ and CCS codes $c \in \mathcal{M}_c$. The hierarchical relationships among these concepts are derived from two sources: (1) Ontology-driven hierarchy graph $\mathcal{G}_m = (\mathcal{V}_m, \mathcal{E}_m)$, where $\mathcal{V}_m$ denotes the set of medical concepts (i.e., diagnoses and CCS codes) and $\mathcal{E}_m$ represents the parent-child relationships. With $\mathcal{G}_m$, we obtain diagnosis box $\mathbf{b}_d$ and CCS code box $\mathbf{b}_c$. (2) EHR-driven hierarchy graph $\mathcal{G}_v = (\mathcal{V}_v, \mathcal{E}_v)$, where $\mathcal{V}_v$ includes all visits and their associated medical concepts, and $\mathcal{E}_v$ captures the relationships between visits and their constituent concepts. With $\mathcal{G}_v$, we obtain visit box $\mathbf{b}_{v_t}$.

The patient's overall representation is then modeled as a box embedding $\mathbf{b}_p$ by aggregating the sequence of visits.

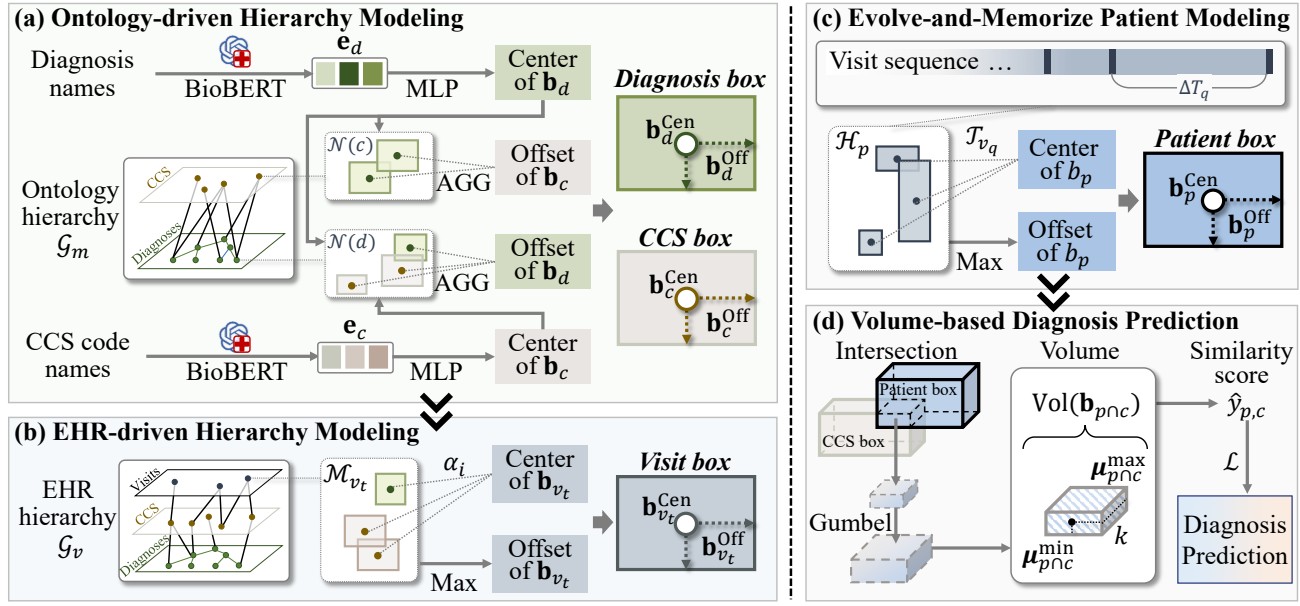

*Figure 2.* The overall framework of our proposed BoxLM.

The prediction is based on the intersection volume-based similarity score $\hat{y}_{p,c}$ between the patient box $\mathbf{b}_p$ and the boxes of candidate CCS codes $\mathbf{b}_c$.

We summarize the main modules of the BoxLM framework in Figure 2 to provide an overview. BoxLM consists of two components. The first component, *Structure-Semantic Fusion via Box Embeddings*, integrates semantic embeddings from PLMs with structural relationships derived from both ontology and EHR data. The second component, *Box-aware Diagnosis Prediction*, models the temporal progression of patient visits and predicts future CCS codes based on the intersection volume between patient and CCS code boxes.

### 2.2. Structure-Semantic Fusion via Box Embeddings

To effectively capture both semantic and structural relationships in medical concepts, we introduce a box-aware language model that models entities as high-dimensional hyperrectangles. This method integrates semantic embeddings from PLMs with structural relationships, directly producing interpretable box representations at the conceptual level. We now proceed to describe how BoxLM is designed to capture hierarchical structures from both ontology and EHR data.

#### 2.2.1. ONTOLOGY-DRIVEN HIERARCHY MODELING

Box embeddings model medical concepts as high-dimensional hyperrectangles, where the center and offset of each box explicitly represent the semantic meaning and hierarchical relationships of these concepts, respectively. In this part, we detail how these box embeddings are computed

during fusion and used to encode hierarchical structures.

**Definition 1 (Box Embedding).** A box is defined as: $\mathbf{b}_i = (\mathbf{b}_i^{\text{Cen}}, \mathbf{b}_i^{\text{Off}})$, where $\mathbf{b}_i^{\text{Cen}} \in \mathbb{R}^{dim}$ and $\mathbf{b}_i^{\text{Off}} \in \mathbb{R}_+^{dim}$ represent the center and offset of the box, respectively.

To preserve the semantic information and adapt it for clinical tasks, we compute the box center using PLM-based semantic embeddings (shown in Figure 2(a)). Specifically, we retrieve the $\mathbf{e}_d$ and $\mathbf{e}_c$ embeddings for medical concepts from BioBERT (Lee et al., 2020) by using their clinical names (e.g., disease names and CCS names) as indices. These embeddings are then passed through a learnable Multi-Layer Perceptron (MLP) to reduce dimensionality and obtain the box center $\mathbf{b}_{\{d,c\}}^{\text{Cen}}$ as follows:

$$\mathbf{b}_d^{\text{Cen}} = \text{MLP}(\mathbf{e}_d), \quad \mathbf{b}_c^{\text{Cen}} = \text{MLP}(\mathbf{e}_c). \tag{1}$$

While PLMs effectively capture the semantic meaning of medical concepts through box centers, they do not inherently encode the parent-child relationships found in medical ontologies. To address this, we propose to leverage clinical knowledge to calculate offset, so as to represent the relative position of concepts within a hierarchy. For instance, the offset for a parent concept, such as *cardiovascular disease*, should be larger than that of its child concept, such as *hypertension*, reflecting the broader nature of the parent concept. This approach ensures that hierarchical relationships are explicitly captured within the box embedding.

To encode hierarchical relationships into offset embeddings, we propose a directed graph-based convolution mechanism

operating over a hierarchical graph. Specifically, in the ontology-driven hierarchy graph $\mathcal{G}_m = (\mathcal{V}_m, \mathcal{E}_m)$, each medical concept $m \in \mathcal{V}_m$ (e.g., *cardiovascular disease* or a CCS code) is initialized with their center embedding $\mathbf{b}_{\{d,c\}}^{\text{Cen}}$, and we adopt relation-aware graph convolutional networks to compute their offset embedding $\mathbf{b}_{\{d,c\}}^{\text{Off}}$, which jointly embeds both nodes and relations in a directed graph. The offset embedding is computed as:

$$\begin{aligned} \mathbf{b}_d^{\text{Off}} &= \text{AGG}\left(\{\phi(\mathbf{b}_n^{\text{Cen}}, \boldsymbol{W}_r) \mid (n,r) \in \mathcal{N}(d)\}\right), \\ \mathbf{b}_c^{\text{Off}} &= \text{AGG}\left(\{\phi(\mathbf{b}_d^{\text{Cen}}, \boldsymbol{W}_r) \mid (d,r) \in \mathcal{N}(c)\}\right), \end{aligned} \quad (2)$$

where $\text{AGG}(\cdot)$ is an aggregation function (e.g., summation or mean) that combines the information from neighboring concepts. $\mathcal{N}(d)$ represents the set of neighboring concepts of diagnosis $d$, and $\mathcal{N}(c)$ denotes the set of diagnosis concepts associated with the CCS code $c$. $\phi(\cdot, \boldsymbol{W}_r)$ is a composition function that combines neighboring concept with the relation $r$, $\boldsymbol{W}_r \in \mathbb{R}^{dim \times dim}$ is the learnable weight, and $r$ encompasses two types of directed edges (i.e., parent-to-child and child-to-parent relationships).

### 2.2.2. EHR-DRIVEN HIERARCHY MODELING

In EHR data, each patient visit is represented as a set of medical concepts, including diagnoses and CCS codes. The aggregation of diagnoses and CCS codes into a visit naturally forms a hierarchical structure $\mathcal{G}_v$, where the visit serves as a parent node encompassing its associated diagnoses and CCS codes as child nodes. Unlike ontology-driven hierarchies, which are predefined by medical ontologies, EHR-driven hierarchies emerge from the clinical context of patient visits.

To model this structure, we represent each visit $v_t$ as a box embedding $\mathbf{b}_{v_t} = (\mathbf{b}_{v_t}^{\text{Cen}}, \mathbf{b}_{v_t}^{\text{Off}})$, where the center $\mathbf{b}_{v_t}^{\text{Cen}}$ captures the semantic meaning of the visit, and the offset $\mathbf{b}_{v_t}^{\text{Off}}$ reflects the variability introduced by the diversity of its associated medical concepts (shown in Figure 2(b)). The center of the visit's box embedding is computed as a weighted aggregation of the box centers of its associated diagnoses and CCS codes:

$$\mathbf{b}_{v_t}^{\text{Cen}} = \sum_{i \in \mathcal{M}_{v_t}} \alpha_i \cdot \mathbf{b}_i^{\text{Cen}}, \quad (3)$$

where $\mathbf{b}_i^{\text{Cen}}$ represents the center of the box for each diagnosis or CCS code $i \in \mathcal{M}_{v_t}$, and $\alpha_i$ is the attention score that captures the relevance of each medical concept to the visit. The attention score is computed as:

$$\alpha_i = \frac{\exp\left(\text{MLP}_E(\mathbf{b}_i^{\text{Cen}})\right)}{\sum_{j \in \mathcal{M}_{v_t}} \exp\left(\text{MLP}_E(\mathbf{b}_j^{\text{Cen}})\right)}. \quad (4)$$

The offset of the visit's box embedding, $\mathbf{b}_{v_t}^{\text{Off}}$, represents the range of the visit's representation, which is influenced by the diversity of its associated medical concepts. A visit with a narrow range of diagnoses and CCS codes will have a smaller offset, while a visit with diverse or conflicting medical concepts will have a larger offset. The offset is computed as the maximum of the offsets of its associated medical concepts:

$$\mathbf{b}_{v_t}^{\text{Off}} = \max(\{\mathbf{b}_i^{\text{Off}} \mid i \in \mathcal{M}_{v_t}\}), \quad (5)$$

where $\mathbf{b}_i^{\text{Off}}$ is the offset for each medical concept $i$ associated with the visit $v_t$ and $\max(\cdot)$ is an element-wise maximum operation.

### 2.3. Box-aware Diagnosis Prediction

In clinical practice, a patient's medical history is represented as a sequence of visits over time, where each visit contains multiple medical concepts such as diagnoses and CCS codes. As a patient's condition evolves, visits often exhibit overlaps in medical concepts, reflecting recurring conditions or related diagnoses. Capturing these overlaps while maintaining an awareness of the temporal progression is essential for accurate diagnosis prediction. Traditional point-based embeddings struggle to model such complex relationships effectively, as they fail to represent the inherent hierarchical structure and semantic relationships of medical concepts. This motivates the use of box embeddings, which naturally model overlaps, hierarchies, and semantic relationships through high-dimensional hyperrectangles, providing a more expressive and interpretable representation for diagnosis prediction.

### 2.3.1. EVOLVE-AND-MEMORIZE PATIENT MODELING

To effectively model the dynamic nature of patient visits while preserving structural and semantic relationships among medical concepts, we propose an evolve-and-memorize patient box learning mechanism (shown in Figure 2(c)). This mechanism leverages the geometric properties of boxes, where the center evolves over time to capture the temporal progression of patient visits $\mathcal{H}_p$, and the offset memorizes the structural and semantic boundaries of the patient's medical history. Specifically, the evolve component dynamically updates the patient representation by weighting historical visits based on their time intervals, while the memorize component aggregates information from all historical visits to preserve hierarchical and semantic relationships. By integrating these two perspectives, our BoxLM achieves a robust and interpretable understanding of patient data:

$$\begin{aligned} \mathbf{b}_p^{\text{Cen}} &= \sum_{v_q \in \mathcal{H}_p} \mathcal{T}_{v_q} \cdot \mathbf{b}_{v_q}^{\text{Cen}}, \\ \mathbf{b}_p^{\text{Off}} &= \max(\{\mathbf{b}_{v_q}^{\text{Off}} \mid v_q \in \mathcal{H}_p\}), \end{aligned} \quad (6)$$

where the influence of historical visits $\mathcal{T}_{v_q}$ is calculated as:

$$\mathcal{T}_{v_q} = \frac{\exp\left(\text{MLP}_T(\Delta T_q)\right)}{\sum_{j=1}^{|\mathcal{H}_p|} \exp\left(\text{MLP}_T(\Delta T_j)\right)}, \tag{7}$$

where $\Delta T_q$ is the time interval between visit $v_q$ and $v_t$.

### 2.3.2. VOLUME-BASED DIAGNOSIS PREDICTION

Given the high-dimensional and structural nature of box embeddings, we propose to leverage the intersection volume between the patient box and CCS code boxes as a measure of their similarity (shown in Figure 2(d)). By quantifying the overlap between boxes, our method provides intuitive insights into the relationships between medical concepts, thereby improving the accuracy and interpretability of model predictions.

**Definition 2 (Box Minimum/Maximum Corner).** The minimum and maximum corners of the box $\mathbf{b}_i$ are given by $\mathbf{b}_i^{\min} = \mathbf{b}_i^{\text{Cen}} - \mathbf{b}_i^{\text{Off}}$ and $\mathbf{b}_i^{\max} = \mathbf{b}_i^{\text{Cen}} + \mathbf{b}_i^{\text{Off}}$.

**Definition 3 (Box Volume).** The volume of box $\mathbf{b}_i$ is calculated by $\text{Vol}(\mathbf{b}_i) = \prod_{k=1}^{dim} \max(\mathbf{b}_{i,k}^{\max} - \mathbf{b}_{i,k}^{\min}, 0)$, where $k$ is the indicator of dimension.

**Definition 4 (Box Intersection).** If there is an overlap between boxes $\mathbf{b}_i$ and $\mathbf{b}_j$, their intersection box is denoted as $\mathbf{b}_{i\cap j} = \mathbf{b}_i \cap \mathbf{b}_j$, where the minimum and maximum corners of the intersection box are $\mathbf{b}_{i\cap j}^{\min} = \max(\mathbf{b}_i^{\min}, \mathbf{b}_j^{\min})$ and $\mathbf{b}_{i\cap j}^{\max} = \min(\mathbf{b}_i^{\max}, \mathbf{b}_j^{\max})$.

Despite the advantages of using box volume-based similarity, a common challenge is the potential for gradient vanishing when the boxes do not overlap (Lin et al., 2024; Liang et al., 2023; Jiang et al., 2023). To address this issue, we draw inspiration from GumbelBox (Dasgupta et al., 2020) and introduce an effective approach that integrates the Gumbel distribution $g(x; \mu, \beta) = \frac{1}{\beta} \exp\left(-\frac{x-\mu}{\beta} e^{-\frac{x-\mu}{\beta}}\right)$ to calculate the intersection box volume, ensuring non-zero gradients and stable optimization.

Specifically, the minimum and maximum corners of the intersection box between the patient box and CCS code boxes are derived using Gumbel distributions as follows:

$$
\begin{aligned}
\mathbf{b}_{p\cap c}^{\min} &= \max(\mathbf{b}_p^{\min}, \mathbf{b}_c^{\min}) \sim \text{MaxGumbel}(\boldsymbol{\mu}_{p\cap c}^{\min}, \beta), \\
\mathbf{b}_{p\cap c}^{\max} &= \min(\mathbf{b}_p^{\max}, \mathbf{b}_c^{\max}) \sim \text{MinGumbel}(\boldsymbol{\mu}_{p\cap c}^{\max}, \beta), \\
\boldsymbol{\mu}_{p\cap c}^{\min} &= \beta \ln\left(\exp^{\mathbf{b}_p^{\min}/\beta} + \exp^{\mathbf{b}_c^{\min}/\beta}\right), \\
\boldsymbol{\mu}_{p\cap c}^{\max} &= -\beta \ln\left(\exp^{-\mathbf{b}_p^{\max}/\beta} + \exp^{-\mathbf{b}_c^{\max}/\beta}\right),
\end{aligned}
\tag{8}
$$

where MaxGumbel/MinGumbel are the max/min stable Gumbel distribution with location $\boldsymbol{\mu}$ and scale $\beta$.

The intersection box expected volume is computed by considering each dimension $k$ independently, as the volume of

*Table 1.* Statistics of the datasets used in our experiments.

| Dataset | MIMIC-III | MIMIC-IV |
|---|---|---|
| # of patients | 5,449 | 79,393 |
| # of visits | 14,141 | 329,597 |
| Avg. # visits per patient | 2.60 | 4.15 |
| Max # visits per patient | 29 | 169 |
| # of unique diagnoses | 3,874 | 37,917 |
| # of CCS codes | 285 | 842 |

a high-dimensional box is the product of its side lengths along each dimension. This allows us to approximate the volume as follows:

$$
\begin{aligned}
\text{Vol}(\mathbf{b}_{p\cap c}) &= \prod_{k=1}^{dim} 2\beta K_0\left(2\exp^{-\left(\boldsymbol{\mu}_{p\cap c,k}^{\max} - \boldsymbol{\mu}_{p\cap c,k}^{\min}\right)/2\beta}\right), \\
&\approx \prod_{k=1}^{dim} \beta \log\left(1 + \exp^{-\left(\boldsymbol{\mu}_{p\cap c,k}^{\max} - \boldsymbol{\mu}_{p\cap c,k}^{\min}\right)/\beta - 2\gamma}\right),
\end{aligned}
\tag{9}
$$

where $K_0$ is the modified Bessel function of the second kind of order zero and $\gamma$ is the Euler-Mascheroni constant. This formulation leverages noise ensembles over a large collection of boxes, enabling the model to escape plateaus, alleviate gradient vanishing, and stabilize the optimization process (Huang et al., 2023). The detailed proof of this formulation is provided in Appendix B.

The box volume-based similarity score for the patient box $\mathbf{b}_p$ and CCS code box $\mathbf{b}_c$ is computed as:

$$\hat{\boldsymbol{y}}_{p,c} = \log(\text{Vol}(\mathbf{b}_{p\cap c})), \tag{10}$$

where the logarithm function $\log$ further prevents the gradient vanishing issue.

To model the multi-label nature of diagnosis prediction, we adopt Binary Cross-Entropy Loss, ensuring robust training by providing a probabilistic interpretation of the similarity scores. The objective function is defined as follows:

$$\mathcal{L} = -\sum_{c=1}^{|\mathcal{M}_c|} \boldsymbol{y}_{p,c} \log(\hat{\boldsymbol{y}}_{p,c}) + (1 - \boldsymbol{y}_{p,c}) \log(1 - \hat{\boldsymbol{y}}_{p,c}), \tag{11}$$

where $\boldsymbol{y}_{p,c}$ is the ground truth label, and $\hat{\boldsymbol{y}}_{p,c}$ is the predicted probability for patient $p$ and CCS code $c$, which are normalized using the softmax function to ensure they represent valid probabilities for multi-label classification.

## 3. Experiments

### 3.1. Experimental Setup

**Datasets and Evaluation Protocols.** We use two real-world EHR datasets to verify the effectiveness of compared meth-

*Table 2.* Experimental results for diagnosis prediction (%) on the MIMIC-III and MIMIC-IV datasets with 5% training data. The best performances are highlighted in **boldface** and the second runners are underlined. *Improv.* denotes the relative improvements of our proposed BoxLM over the second runners.

| Dataset | MIMIC-III | | | | MIMIC-IV | | | |
|---|---|---|---|---|---|---|---|---|
| | Visit-Level | | Code-Level | | Visit-Level | | Code-Level | |
| Metric | P@10 | P@20 | Acc@10 | Acc@20 | P@10 | P@20 | Acc@10 | Acc@20 |
| DoctorAI | $35.72_{\pm 0.16}$ | $43.70_{\pm 0.20}$ | $26.31_{\pm 0.19}$ | $42.64_{\pm 0.21}$ | $30.65_{\pm 0.05}$ | $37.29_{\pm 0.07}$ | $22.69_{\pm 0.06}$ | $34.83_{\pm 0.05}$ |
| RETAIN | $34.73_{\pm 0.18}$ | $43.10_{\pm 0.18}$ | $25.73_{\pm 0.17}$ | $42.00_{\pm 0.19}$ | $33.47_{\pm 0.04}$ | $40.56_{\pm 0.05}$ | $24.52_{\pm 0.03}$ | $37.54_{\pm 0.06}$ |
| StageNet | $36.02_{\pm 0.19}$ | $43.91_{\pm 0.21}$ | $26.47_{\pm 0.16}$ | $42.74_{\pm 0.13}$ | $31.46_{\pm 0.07}$ | $38.02_{\pm 0.08}$ | $22.94_{\pm 0.06}$ | $34.84_{\pm 0.07}$ |
| TRANS | $36.64_{\pm 0.22}$ | $44.85_{\pm 0.23}$ | $26.96_{\pm 0.19}$ | $43.35_{\pm 0.22}$ | $28.70_{\pm 0.05}$ | $35.36_{\pm 0.05}$ | $21.64_{\pm 0.06}$ | $33.31_{\pm 0.04}$ |
| KAME | $35.01_{\pm 0.17}$ | $43.17_{\pm 0.20}$ | $25.95_{\pm 0.21}$ | $42.16_{\pm 0.18}$ | $30.54_{\pm 0.08}$ | $37.16_{\pm 0.11}$ | $22.29_{\pm 0.09}$ | $34.10_{\pm 0.08}$ |
| CGL | $35.54_{\pm 0.22}$ | $43.78_{\pm 0.25}$ | $26.15_{\pm 0.19}$ | $42.49_{\pm 0.20}$ | $31.72_{\pm 0.13}$ | $38.51_{\pm 0.09}$ | $23.14_{\pm 0.11}$ | $35.76_{\pm 0.12}$ |
| HiTANet | $36.23_{\pm 0.15}$ | $44.32_{\pm 0.19}$ | $26.56_{\pm 0.17}$ | $43.02_{\pm 0.16}$ | $28.83_{\pm 0.09}$ | $35.40_{\pm 0.12}$ | $21.75_{\pm 0.12}$ | $33.39_{\pm 0.10}$ |
| BoxCare | $38.21_{\pm 0.17}$ | $45.31_{\pm 0.16}$ | $28.12_{\pm 0.19}$ | $43.84_{\pm 0.21}$ | $35.13_{\pm 0.07}$ | $41.94_{\pm 0.10}$ | $25.58_{\pm 0.09}$ | $38.80_{\pm 0.09}$ |
| BERT | $21.77_{\pm 0.22}$ | $32.19_{\pm 0.23}$ | $16.14_{\pm 0.22}$ | $31.88_{\pm 0.19}$ | $12.92_{\pm 0.12}$ | $20.24_{\pm 0.13}$ | $10.16_{\pm 0.12}$ | $19.27_{\pm 0.11}$ |
| BERT* | $23.81_{\pm 0.19}$ | $32.26_{\pm 0.24}$ | $17.84_{\pm 0.25}$ | $32.15_{\pm 0.25}$ | $13.16_{\pm 0.10}$ | $20.85_{\pm 0.12}$ | $10.95_{\pm 0.11}$ | $19.59_{\pm 0.14}$ |
| BioBERT | $34.17_{\pm 0.21}$ | $43.31_{\pm 0.26}$ | $25.35_{\pm 0.19}$ | $42.06_{\pm 0.20}$ | $18.81_{\pm 0.09}$ | $27.37_{\pm 0.12}$ | $15.32_{\pm 0.10}$ | $25.35_{\pm 0.12}$ |
| BioBERT* | $34.42_{\pm 0.25}$ | $43.44_{\pm 0.29}$ | $25.49_{\pm 0.18}$ | $42.27_{\pm 0.21}$ | $18.95_{\pm 0.11}$ | $28.08_{\pm 0.14}$ | $15.44_{\pm 0.13}$ | $25.61_{\pm 0.13}$ |
| VecoCare | $35.27_{\pm 0.20}$ | $43.54_{\pm 0.19}$ | $25.98_{\pm 0.21}$ | $42.30_{\pm 0.23}$ | $29.17_{\pm 0.10}$ | $35.84_{\pm 0.12}$ | $22.06_{\pm 0.09}$ | $33.89_{\pm 0.12}$ |
| BoxLM | $\mathbf{43.88_{\pm 0.23}}$ | $\mathbf{51.62_{\pm 0.25}}$ | $\mathbf{31.74_{\pm 0.21}}$ | $\mathbf{48.74_{\pm 0.19}}$ | $\mathbf{42.04_{\pm 0.04}}$ | $\mathbf{49.65_{\pm 0.08}}$ | $\mathbf{29.94_{\pm 0.06}}$ | $\mathbf{44.52_{\pm 0.08}}$ |
| *Improv.* | 14.84% | 13.93% | 12.87% | 11.18% | 19.67% | 18.38% | 17.04% | 14.74% |

ods, i.e., MIMIC-III (Johnson et al., 2016) and MIMIC-IV (Johnson et al., 2018). Following (Ma et al., 2018; Chen et al., 2024), we chose patients who made at least two visits for both datasets and predicted the CCS codes for the next visit of patients. The statistics are summarized in Table 1. For evaluation metrics, we use visit-level Precision@k (P@k) and code-level Accuracy@k (Acc@k) from coarse-grained and fine-grained perspectives, which are consistent with (Ma et al., 2018; Zhang et al., 2020; Wang et al., 2023; Chen et al., 2024). As suggested in (Choi et al., 2016a; Lu et al., 2021; Lv et al., 2024), we also report Recall@k (see Appendix G) for more comprehensive evaluation of diagnosis prediction. Details on evaluation metrics are provided in Appendix E.

**Baselines.** To comprehensively evaluate our proposed BoxLM, we adopt 13 representative state-of-the-art methods as baselines for comparison from three main perspectives: (1) **temporal-aware methods**: DoctorAI (Choi et al., 2016a), RETAIN (Choi et al., 2016b), StageNet (Gao et al., 2020), and TRANS (Chen et al., 2024); (2) **hierarchy-aware methods**: KAME (Ma et al., 2018), CGL (Lu et al., 2021), HiTANet (Luo et al., 2020), and BoxCare (Lv et al., 2024); (3) **semantic-aware methods**: BERT (Kenton & Toutanova, 2019), BERT*, BioBERT (Lee et al., 2020), BioBERT*, and VecoCare (Xu et al., 2023b). Here, the superscript * indicates methods that incorporate patients' historical visits during training. Additional details about these baselines are provided in Appendix F.

**Implementation Details.** Both datasets are split into training, validation, and test sets at a ratio of 7:1:2, with pa-

tients as the unit of segmentation, consistent with (Lv et al., 2024). We optimize the compared baselines with the standard Adam optimizer and carefully tune their hyperparameters as suggested in the original papers. In particular, we set the embedding dimension $dim$ as 16 and the scale of Gumbel distribution $\beta$ as 0.2. We evaluate each model with the 5-fold cross-validation strategy. Both the mean and standard deviation of test performance are reported. All experiments are conducted using two NVIDIA GTX 3090 Ti GPUs. The full code for this work is available[1].

### 3.2. Overall Diagnosis Prediction Results

In this section, we assess the effectiveness of our BoxLM for diagnosis prediction under the few-shot and varying ratios of training data scenarios, respectively.

#### 3.2.1. FEW-SHOT DIAGNOSIS PREDICTION

To comprehensively evaluate the performance of our proposed BoxLM, we compare it against 13 state-of-the-art baselines from three main perspectives: temporal-aware, hierarchy-aware, and semantic-aware methods. The results, as shown in Table 2, demonstrate that BoxLM consistently outperforms all baselines across both the MIMIC-III and MIMIC-IV datasets.

**Comparison with hierarchy-aware models.** Among the baselines, BoxCare stands out as a strong hierarchy-aware method that leverages box embeddings to model structural relationships among diagnoses. BoxCare primarily focuses

---

[1]https://github.com/Melinda315/BoxLM

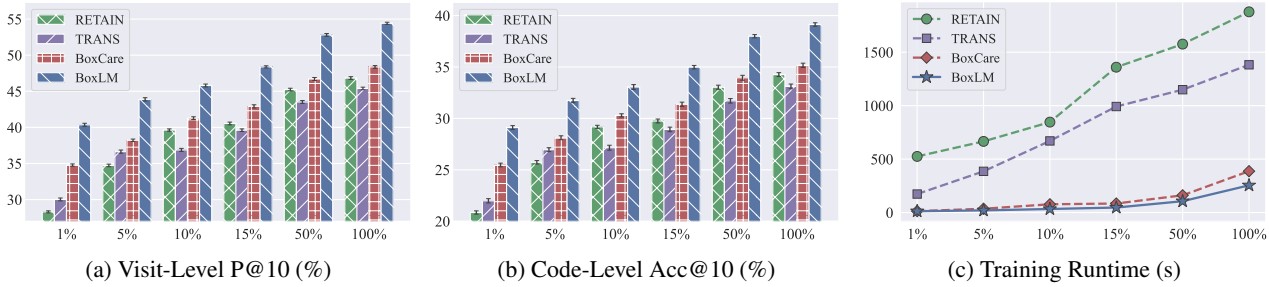

*Figure 3.* Experimental results for diagnosis prediction on the MIMIC-III dataset with varying ratios of training data.

on ontology-driven hierarchies (e.g., ICD codes), which capture standardized relationships between medical concepts. However, it does not fully incorporate EHR-driven hierarchies, which reflect real-world patient visit patterns and medical concept co-occurrences. In contrast, BoxLM explicitly integrates knowledge from pre-trained language models with both ontology-driven and EHR-driven hierarchies. As a result, BoxLM outperforms BoxCare by an average of 16.70% at the visit level and 13.96% at the code level across both datasets, demonstrating its superior ability to unify structural and semantic information of medical concepts for accurate diagnosis prediction.

**Comparison with temporal-aware models.** One representative baseline is RETAIN, which employs RNNs with reverse time attention mechanisms to model the temporal dynamics of patient visits in EHRs. Another notable baseline is TRANS, a temporal graphic method, that leverages transformer-based architectures to model the sequential nature of patient visits. Compared with RETAIN and TRANS, BoxLM combines temporal modeling with a rich representation of medical concepts derived from both EHR data and ontological sources through our structure-semantic fusion mechanism. This integration is particularly evident in BoxLM's superior performance on the MIMIC-IV dataset with more medical entities (shown in Table 1), surpassing TRANS by up to 46.48% in P@10 on MIMIC-IV.

**Comparison with semantic-aware models.** Semantic-aware methods, such as VecoCare and BioBERT, primarily rely on vector-based representations to encode medical concepts. These methods focus on capturing the semantic meaning of individual concepts. For example, BioBERT enhances semantic understanding through domain-specific pre-training, while VecoCare uses PLM embeddings to represent medical concepts. However, their vector-based design inherently limits their ability to model complex hierarchical relationships. In contrast, BoxLM models medical concepts as high-dimensional geometric structures (hyperrectangles) to unify structural and semantic information. This enables BoxLM to achieve an average improvement of 28.76% over semantic-aware methods on both datasets.

### 3.2.2. VARYING RATIOS OF TRAINING DATA

To investigate the impact of training data size on model performance, we conduct experiments on the MIMIC-III dataset with varying ratios of training data, e.g., 1%, 5%, 10%, 15%, 50%, and 100% (shown in Figure 3).

In general, BoxLM significantly outperforms all baselines across all data ratios, demonstrating its ability to generalize even with limited training data. At both the visit and code levels (Figure 3(a) and Figure 3(b)), baselines show improvements as the training data ratio increases, but their performance gains are modest compared to BoxLM. For instance, BoxCare, the second-best method, improves a P@10 at the visit level from 34.78% to 38.21%, as the training data increases from 1% to 5%. However, even with 100% training data, BoxCare's performance in P@10 remains slightly below BoxLM's performance with only 15% training data.

Additionally, the runtime analysis (Figure 3(c)) further highlights the efficiency of BoxLM. Notably, RETAIN and TRANS exhibit significantly longer training times, where RETAIN is based on the reverse time attention mechanisms and TRANS relies on temporal graphical representations. In contrast, BoxLM achieves superior efficiency by leveraging box embeddings to unify structural and semantic representations in a compact and interpretable manner. This design not only reduces computational overhead but also enables faster convergence during training. Compared to BoxCare, which also utilizes box embeddings, BoxLM introduces the structure-semantic fusion mechanism and the evolve-and-memorize patient box learning mechanism. These innovations employ external semantic knowledge from PLM to further enhance efficiency. Moreover, they allow BoxLM to model temporal dynamics and hierarchical relationships effectively with an average 36.57% lower runtime. More experimental results and analyses on MIMIC-IV can be found in Appendix H.

### 3.3. Analysis of Our Framework

**Ablation Studies.** To better understand our proposed techniques, i.e., Ontology-based hierarchy modeling (Onto),

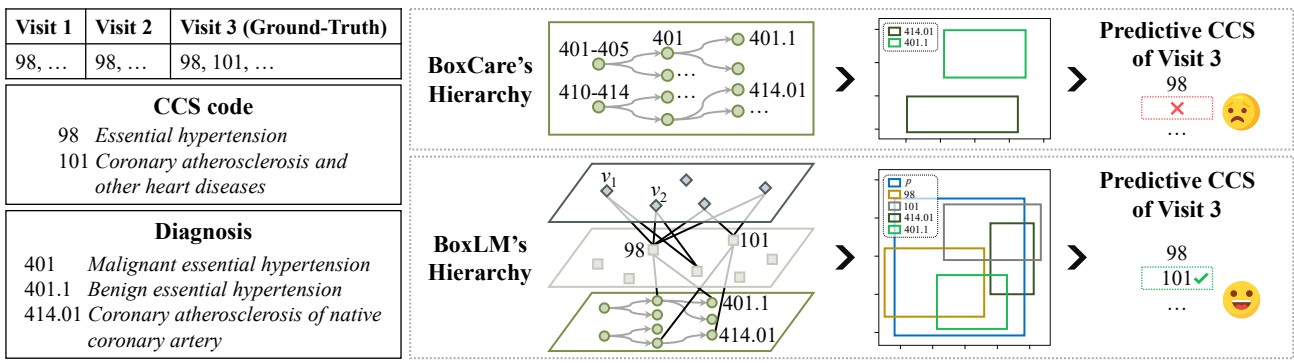

*Figure 4.* An illustrative example of diagnosis prediction for patient $p$ on MIMIC-III generated by BoxCare and BoxLM, respectively.

*Table 3.* Ablation results (%) of our BoxLM on two EHR datasets.

| Metric | Visit-Level | | Code-Level | |
|---|---|---|---|---|
| | P@10 | P@20 | Acc@10 | Acc@20 |
| MIMIC-III | | | | |
| Base | $34.42_{\pm0.25}$ | $43.44_{\pm0.29}$ | $25.49_{\pm0.18}$ | $42.27_{\pm0.21}$ |
| + Onto | $37.07_{\pm0.10}$ | $44.69_{\pm0.15}$ | $26.83_{\pm0.13}$ | $43.18_{\pm0.18}$ |
| + EHR | $41.40_{\pm0.24}$ | $48.74_{\pm0.22}$ | $30.05_{\pm0.13}$ | $46.37_{\pm0.10}$ |
| BoxLM | $43.88_{\pm0.23}$ | $51.62_{\pm0.25}$ | $31.74_{\pm0.21}$ | $48.74_{\pm0.19}$ |
| MIMIC-IV | | | | |
| Base | $18.95_{\pm0.11}$ | $28.08_{\pm0.14}$ | $15.44_{\pm0.13}$ | $25.61_{\pm0.13}$ |
| + Onto | $32.14_{\pm0.03}$ | $38.63_{\pm0.06}$ | $23.29_{\pm0.03}$ | $34.71_{\pm0.04}$ |
| + EHR | $40.28_{\pm0.05}$ | $47.94_{\pm0.04}$ | $28.73_{\pm0.04}$ | $43.09_{\pm0.07}$ |
| BoxLM | $42.04_{\pm0.04}$ | $49.65_{\pm0.08}$ | $29.94_{\pm0.06}$ | $44.52_{\pm0.08}$ |

EHR-based hierarchy modeling (EHR), and evolve-and-memorize patient modeling, we study BoxLM on two EHR datasets as follows:

(1) Base model leverages BioBERT embeddings for medical concepts without any structure modeling, where visits are represented as single points; (2) "+ Onto" introduces box embeddings for CCS codes and diagnoses with our ontology-based hierarchy modeling (cf., Section 2.2.1). Visit representations are constructed by summing the centers and offsets of associated CCS code and diagnosis boxes, followed by concatenation rather than using box representations; (3) "+ EHR" extends to visit-level box embeddings with our EHR-based hierarchy modeling (cf., Section 2.2.2). Patient are vector embeddings similar to the visit embeddings in the "+ Onto" setup; (4) BoxLM models diagnoses, CCS codes, visits, and patients as box embeddings, and employs volume-based similarity for diagnosis prediction.

From Table 3, we have the following observations:

Incorporating box embeddings for hierarchical modeling achieves 7.70% gains with P@10 on MIMIC-III, with even more significant gains on MIMIC-IV (i.e., 69.60%). Such results highlight that datasets with more medical entities (e.g., MIMIC-IV) benefit greatly from modeling ontology-driven hierarchy structures, as they effectively address the complexity of overlapping and related medical concepts.

Rather than representing visits as single points, modeling visits as box embeddings by capturing EHR-driven hierarchy achieves further improvement, with up to 25.33% gains in P@10 on MIMIC-IV. By capturing the relationships between visits and their associated medical concepts (e.g., diagnoses), BoxLM provides a more robust and interpretable framework, especially in datasets with a larger number of visits and medical entities.

By extending box embeddings to patient-level representations, we are able to effectively measure the overlap and complex relationships (e.g., structure, semantic, and temporal relations) among medical entities, leading to more precise diagnosis predictions. These results also affirm that our evolve-and-memorize patient box learning mechanism captures the evolving nature of patient visits.

**Hyperparameter Studies.** We evaluate how three hyperparameters (i.e., the box embedding dimension $dim$, the Gumbel distribution scale $\beta$, and the box volume calculation) impact the performance and clarify how to set them. The detailed results are shown in Appendix I. For box embedding dimension $dim$, the results show that increasing $dim$ slowly enhances model performance for both datasets. For a fair comparison with baselines, we set $dim = 16$. For the Gumbel distribution scale $\beta$, a larger $\beta$ makes the distribution closer to uniform, while a smaller $\beta$ causes its probability density function to approach a hinge function, leading the random variable to degenerate into a constant (Lin et al., 2024). In this paper, we set $\beta = 0.2$, balancing model accuracy with the distinctiveness among boxes. For the box volume calculation, we compare the soft volume calculation (Li et al., 2019) with our used Bessel volume calculation (cf., Eq. (9)). They both effectively mitigate the training difficulties that arise when disjoint boxes should overlap.

**Case Studies.** To highlight the advantages of BoxLM in diagnosis prediction, we provide a real case study from the MIMIC-III dataset. Figure 4 contrasts the predictive results

of visit 3 for patient $p$ from BoxCare and `BoxLM`.

For patient $p$'s diagnosed with *essential hypertension* during the first two visits (i.e., $v_1$ and $v_2$), BoxCare relies on ontology-driven hierarchy for diagnosis boxes and models patients using vector embeddings. Consequently, it cannot capture the co-occurrence patterns present in EHR data, leading to inaccurate diagnosis predictions. In contrast, our proposed `BoxLM` make full use of ontology-driven and EHR-driven hierarchy by designing the structure-semantic fusion mechanism and evolve-and-memorize patient box learning mechanism, empowering the model to uncover fine-grained, interpretable associations among medical entities. Therefore, `BoxLM` accurately forecasts the possibility of developing *coronary atherosclerosis and other heart diseases* (with CCS code 101) by calculating the overlap between the accurate patient and CCS boxes—consistent with clinical knowledge that hypertension increases the heart's workload on arteries (Dzau, 1990; Libby et al., 2009), thereby elevating the risk of atherosclerosis. Additionally, we provide a quantitative metric Consistency@$k$ metric to verify the interpretability of `BoxLM` (see Appendix J).

## 4. Related work

**Diagnosis Prediction.** Diagnosis prediction has been extensively studied due to its pivotal role in healthcare (Hendriksen et al., 2013; Yang et al., 2023). A significant category of deep learning-based methods focuses on modeling contextual dependencies within patient visit sequences using architectures like Recurrent Neural Networks (RNNs) and transformers, such as DoctorAI (Choi et al., 2016a), RE-TAIN (Choi et al., 2016b), Dipole (Ma et al., 2017), and TRANS (Chen et al., 2024). Another prominent direction involves integrating external knowledge to enhance representation learning (Choi et al., 2017; Ye et al., 2021; Xu et al., 2023a). For example, hierarchy-aware methods (Lu et al., 2021; Wang et al., 2023; Lv et al., 2024) have demonstrated strong performance by explicitly encoding disease domain knowledge. However, these methods often neglect the rich semantic information among medical concepts, which is valuable for diagnosis prediction modeling the contextual relationships and similarities between medical terms.

**Clinical Language Model.** LMs have significantly advanced clinical tasks by incorporating rich semantic information (Kenton & Toutanova, 2019; Yan & Pei, 2022). For example, BioBERT (Lee et al., 2020) and PubMed-BERT (Gu et al., 2021) utilized biomedical text corpora for pre-training, enabling improved representation of domain-specific medical terminology. VecoCare (Xu et al., 2023b) aggregated semantic information from patients through a dual-channel retrieval mechanism. In recent years, medical-specific LLMs have leveraged large-scale medical datasets to further enhance the contextual understanding of clinical concepts (Kwon et al., 2024; Kim et al., 2024). However, existing LM-based models face challenges in effectively encoding and interpreting hierarchical structures latent in language, which can be helpful for diagnosis prediction.

**Box Embedding.** Box embeddings have emerged as a powerful geometric representation approach for capturing complex hierarchical and relational information. Unlike traditional vector embeddings, box embeddings model entities as high-dimensional hyperrectangles, enabling the representation of inclusion, exclusion, and intersection relationships (Mei et al., 2022). Motivated by the box embeddings, which can clearly capture the complex structural relations, real-world applications such as logical query answering (Ren et al., 2020), knowledge base completion (Huang et al., 2023), taxonomy expansion (Jiang et al., 2023), recommendation (Lin et al., 2024), and diagnosis prediction (Lv et al., 2024) have been proposed. As closest to us, BoxCare (Lv et al., 2024) applied box embeddings to represent disease hierarchies for clinical predictions. However, BoxCare primarily focused on ontology hierarchies without fully integrating the rich semantics and temporal structures inherent in EHRs. For a more comprehensive discussion of related work, please refer to Appendix A.

## 5. Conclusion

In this paper, we propose `BoxLM`, a novel framework that unifies structural and semantic representations of medical concepts for diagnosis prediction. `BoxLM` includes a structure-semantic fusion mechanism that integrates ontology-driven and EHR-driven hierarchies with LM-based semantic embeddings, and a box-aware diagnosis prediction module that models temporal dynamics and quantifies relationships between patient and medical concepts. Extensive experiments demonstrate that `BoxLM` consistently outperforms state-of-the-art baselines, particularly in few-shot learning scenarios, showcasing its practical utility in real-world clinical settings.

For future work, we aim to explore the application of `BoxLM` to other medical tasks (e.g., medication recommendation, medical report generation, and medical image segmentation) and investigate its integration with multimodal data sources (e.g., clinical notes, X-rays, and sensor records) to further enhance its capabilities and application value.

## Disclosure of Generative AI

We acknowledge the use of ChatGPT (OpenAI[2]) to assist in proofreading and improving the structure of this essay. The prompts we used include: "Please check the grammar and suggest improvements for clarity."

---

[2]https://chat.openai.com/

## Acknowledgements

This work was supported in part by the National Natural Science Foundation of China under Grants (No.62302098), Fujian Provincial Natural Science Foundation of China under Grants (2025J01540), Zhejiang Provincial Natural Science Foundation of China under Grants (LQ23F020007), and Zhejiang Provincial Department of Agriculture and Rural Affairs Project under Grants (2024SNJF044). Carl Yang was not supported by any fund from China.

## Impact Statement

This paper presents work whose goal is to advance the field of machine learning in healthcare by proposing BoxLM, a novel framework for diagnosis prediction. BoxLM unifies structural and semantic representations of medical concepts, enabling accurate and interpretable predictions while minimizing the reliance on large-scale patient-specific data. By leveraging anonymized Electronic Health Records (EHRs) and incorporating hierarchical ontologies, BoxLM reduces the risk of privacy infringement and addresses the challenges of data scarcity in clinical settings.

As a decision-support tool, BoxLM is designed to assist healthcare professionals rather than replace them, aiming to reduce their workload and improve diagnostic efficiency. Its strong performance in few-shot learning scenarios makes it particularly valuable in resource-constrained environments, such as rural or underdeveloped areas, where access to extensive patient data is limited. By advancing the integration of structural and semantic knowledge in medical AI, BoxLM contributes to the development of intelligent, accessible, and ethical diagnostic tools, with the potential to improve patient outcomes and healthcare equity.

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

## A. Detailed Related Work

**Traditional Diagnosis Prediction.** Early diagnosis prediction methods primarily relied on traditional machine learning approaches, such as logistic regression and random forest, which struggled to effectively handle the high-dimensional and sparse nature of EHR data (Hendriksen et al., 2013). The advent of deep learning has significantly advanced predictive modeling in diagnosis (Bai et al., 2018; Li et al., 2020). For example, RETAIN (Choi et al., 2016b) employed a two-level attention-based RNN. Dipole (Ma et al., 2017) proposed a bidirectional RNN architecture with three attention mechanisms. TRANS (Chen et al., 2024) modeled the patient's EHR as a temporal heterogeneous graph and utilized time-aware visit nodes to capture patient health status changes. Timeline (Bai et al., 2018) utilized time-aware attention mechanisms in RNNs for health event predictions. Chet (Lu et al., 2022) designed a context-aware dynamic graph learning method to learn disease combinations and disease development schemes. Another prominent direction involves integrating external knowledge graphs to enhance representation learning, such as GRAM (Choi et al., 2017), MedPath (Ye et al., 2021), and SeqCare (Xu et al., 2023a).

**Structure-aware Diagnosis Prediction.** Existing structure-aware models mainly use the hierarchy of medical ontologies(e.g., ICD-9) for diagnosis prediction. For example, HiTANet (Luo et al., 2020) adopted the time-aware Transformer and attention mechanism to capture correlations between visits from local and global views. KAME (Ma et al., 2018) proposed a graph-based attention model to obtain good performance with insufficient data. CGL (Lu et al., 2021) captured structural features of both patients and diseases, integrating attentive text features into a sequential learning process.

**Medical Large Language Models.** With the advent of LLMs, medical artificial intelligence has undergone significant technological advancements and paradigm shifts, demonstrating the potential of LLMs to enhance healthcare delivery and improve patient outcomes (Liu et al., 2024). Starting with BioBERT(Lee et al., 2020) in 2019, the field has seen the development of specialized models for various medical tasks. Key milestones include BioMegatron(Shin et al., 2020) and PubMedBERT(Gu et al., 2021), followed by the introduction of models like ChatDoctor(Li et al., 2023) and ClinicalBERT(Yan & Pei, 2022). The field continues to evolve with advanced models such as MedAgents (Tang et al., 2024), MedREQAL (Vladika et al., 2024), PubMedBERT(Saab et al., 2024), and Health-LLM (Kim et al., 2024), underscoring the ongoing innovation in leveraging LLMs for diverse healthcare applications.

**Geometric Embedding.** Geometric embedding models have attracted significant attention for their ability to preserve the intrinsic geometric structure of data (Lin et al., 2024; Chen et al., 2022). Among them, box embeddings represent entities as high-dimensional hyperrectangles. This makes them particularly effective for tasks involving hierarchical (Onoe et al., 2021), transitive relations (Subramanian & Chakrabarti, 2018), entailments (Li et al., 2019), and uncertainty (Vilnis & McCallum, 2015), thereby inferring missing or incomplete hierarchical links in data. Motivated by the box embeddings, various real-world applications have been proposed, such as taxonomy expansion (Jiang et al., 2023), knowledge base completion (Huang et al., 2023; Abboud et al., 2020), taxonomy expansion (Jiang et al., 2023), recommendation (Lin et al., 2024; Wu et al., 2024), and diagnosis prediction (Lv et al., 2024). Additionally, recent advances extend this idea to more expressive geometric forms such as polygons and hypercubes (Pavlović & Sallinger, 2023; Chen et al., 2022), further enhancing performance in domains like knowledge graphs and recommender systems.

## B. Proof of Intersection Box Expected Volume Formulation

The detailed derivation of $\boldsymbol{\mu}_{p \cap c}^{\min}$ and $\boldsymbol{\mu}_{p \cap c}^{\max}$ is provided in (Dasgupta et al., 2020). According to **Definition 3** in Section 2.3.2, we obtain the intersection box expected volume by calculating the expected length for each dimension as follows:

$$\text{Vol}(\mathbf{b}_{p \cap c}) = \mathbb{E}[\max(\mathbf{b}_{p \cap c}^{\max} - \mathbf{b}_{p \cap c}^{\min}, 0)] = \prod_{k=1}^{dim} 2\beta K_0 \left( 2 \exp^{-(\boldsymbol{\mu}_{p \cap c,k}^{\max} - \boldsymbol{\mu}_{p \cap c,k}^{\min})/2\beta} \right), \tag{12}$$

where $dim$ is the dimension of embeddings and $K_0$ is the modified Bessel function of the second kind of order zero. The proof of this formulation is detailed in (Dasgupta et al., 2020).

Next, we define $f(x) = 2\beta K_0 \left( 2 \exp^{x/2\beta} \right)$. Obviously, the function $f(x)$ is essentially exponential as $x$ increases. The volume function approaches a hinge function as $\beta \to 0$, which leads to numerical stability concerns (Dasgupta et al., 2020). Consequently, we employ the softplus function to approximate $f(x)$:

$$f(x) \approx \beta \log \left( 1 + \exp^{x/\beta - 2\gamma} \right), \tag{13}$$

where $\gamma$ is the Euler-Mascheroni constant. Based on this approximation, the formulation of the intersection box expected volume can be derived as follows:

$$\text{Vol}(\mathbf{b}_{p\cap c}) \approx \prod_{k=1}^{dim} \beta \log\left(1 + \exp^{-\left(\boldsymbol{\mu}_{p\cap c,k}^{\max} - \boldsymbol{\mu}_{p\cap c,k}^{\min}\right)/\beta - 2\gamma}\right). \tag{14}$$

## C. Complexity Analysis of **BoxLM**

We analyze the time complexity of the `BoxLM` framework by four major components: Ontology-driven Hierarchy Modeling, EHR-driven Hierarchy Modeling, Evolve-and-Memorize Patient Modeling, and Volume-based Diagnosis Prediction.

(1) **Ontology-driven Hierarchy Modeling:** In the ontology-driven hierarchy graph $\mathcal{G}_m = (\mathcal{V}_m, \mathcal{E}_m)$, the offset embeddings of medical concepts (i.e., diagnoses and CCS codes) are computed through relation-aware graph convolution networks. The complexity per layer is $\mathcal{O}\left(|\mathcal{E}_m| \times dim^2 + |\mathcal{R}_m| \times dim^2\right)$, where $|\mathcal{E}_m|$ is the number of edges in $\mathcal{G}_m$, $|\mathcal{R}_m|$ is the number of relation types (e.g., parent-child relationship), and $dim$ is the embedding dimension.

(2) **EHR-driven Hierarchy Modeling:** In the EHR-driven hierarchy graph, each visit $v_t$ is associated with a set of medical concepts. We represent the visit's box embedding via an attention-weighted aggregation and a maximum operation. The complexity is $\mathcal{O}(|\mathcal{M}_{v_t}| \times dim)$, where $|\mathcal{M}_{v_t}|$ is the number of medical concepts associated with visit $v_t$.

(3) **Evolve-and-Memorize Patient Modeling:** The patient box aggregates information from all historical visits $\mathcal{H}_p$ via temporal weighting and the complexity is $\mathcal{O}(|\mathcal{H}_p| \times dim)$, where $|\mathcal{H}_p|$ is the number of the patient's historical visits.

(4) **Volume-based Diagnosis Prediction:** In the diagnosis prediction phase, each patient box is compared to $|\mathcal{M}_c|$ candidate CCS boxes via the Gumbel-approximated intersection volume. For $N_p$ patients, the time complexity is $\mathcal{O}(N_p \times |\mathcal{M}_c| \times dim)$, which grows linearly with the embedding $dim$ and is similar to traditional vector-based approaches.

## D. Notations and Corresponding Description

As shown in Table 4, we summarize the key notations used in our `BoxLM` and their corresponding descriptions.

*Table 4.* Notations

| Notation | Description |
|---|---|
| $d, c, v$ | The medical concepts, including diagnosis $d$, CCS code $c$, and visit $v$. |
| $\mathbf{e}_d, \mathbf{e}_c$ | The embedding of medical concepts $d, c$ derived from PLM. |
| $\mathbf{b}_i = (\mathbf{b}_i^{\text{Cen}}, \mathbf{b}_i^{\text{Off}})$ | The box embedding of concept $i$, where $\mathbf{b}_i^{\text{Cen}}$ and $\mathbf{b}_i^{\text{Off}}$ are the center and offset of the box. |
| $\mathbf{b}_i^{\min}$ | The minimum corner of the box $\mathbf{b}_i$. |
| $\mathbf{b}_i^{\max}$ | The maximum corner of the box $\mathbf{b}_i$. |
| $\text{Vol}(\mathbf{b}_i)$ | The volume of the box $\mathbf{b}_i$. |
| $\mathbf{b}_{i\cap j}$ | The intersection box between boxes $\mathbf{b}_i$ and $\mathbf{b}_j$. |
| $\text{MinGumbel}(\boldsymbol{\mu}, \beta)$ | The min stable Gumbel distribution. |
| $\text{MaxGumbel}(\boldsymbol{\mu}, \beta)$ | The max stable Gumbel distribution. |
| $\boldsymbol{\mu}$ | The location vector of the Gumbel distribution. |
| $\beta$ | The scale parameter of the Gumbel distribution. |
| $\mathcal{N}(d), \mathcal{N}(c)$ | The set of neighboring concepts of diagnosis $d$ and CCS code $c$. |
| $\mathcal{M}_d, \mathcal{M}_c$ | The set of medical concepts $d, c$. |
| $\mathcal{M}_{v_t}$ | The set of associated medical concepts of visit $v_t$. |
| $\mathcal{H}_p$ | The temporal progression of patient visits. |

## E. Details of Evaluation Metrics

Following previous studies (Ma et al., 2018; Chen et al., 2024; Choi et al., 2016a; Lu et al., 2021; Lv et al., 2024), three evaluation metrics are adopted for diagnosis prediction: Visit-level Precision@$k$ (P@$k$) measures prediction effectiveness by counting accurate medical code predictions within the top $k$ ranked results, normalized by the minimum of $k$ and the

actual number of categories present in a patient visit. Code-level Accuracy@$k$ (Acc@$k$) quantifies prediction reliability as the proportion of correct diagnoses among all predicted medical codes at the individual code level. Recall@$k$ measures the ratio of true medical codes in the top $k$ predictions by the total number of ground-truth medical codes, reflecting the model's ability to retrieve relevant diagnoses. The value of $k$ ranges from 10 to 20 across both metrics, with higher values indicating better model effectiveness, where visit-level precision evaluates broader performance and code-level accuracy reflects more detailed correctness. Recall mimics the clinical process of differential diagnosis, where physicians identify the most probable conditions for evaluation. The calculation formulas are detailed in (Chen et al., 2024; Choi et al., 2016a).

## F. Details of Compared Baselines

We compare 13 representative state-of-the-art methods as baselines for comparison from three main perspectives:

(1) **Temporal-aware methods** concentrate on modeling temporal dynamics and sequences in patient data. DoctorAI (Choi et al., 2016a) encodes visits into vector representations and then processes them with GRUs to predict diagnoses in future visits. RETAIN (Choi et al., 2016b) employs RNNs that integrate a reverse time attention mechanism to predict patient diagnoses. StageNet (Gao et al., 2020) proposes a stage-aware LSTM module and a stage-adaptive convolutional module for diagnosis prediction. TRANS (Chen et al., 2024) constructs a temporal heterogeneous graph to jointly capture temporal dynamics and structural relationships in EHR data.

(2) **Hierarchy-aware methods** emphasize capturing hierarchical structures within medical data. KAME (Ma et al., 2018) focuses on predicting patients' future health information by incorporating medical ontology knowledge of ICD codes into the sequence model. HiTANet (Luo et al., 2020) proposes a hierarchical time-aware transformer for risk prediction based on EHRs. CGL (Lu et al., 2021) designs a collaborative graph learning model to explore patient-disease interactions and external medical knowledge. BoxCare (Lv et al., 2024) employs box embeddings to model both inclusive and exclusive relations among diseases and ICD-9 codes in EHR data.

(3) **Semantic-aware methods** primarily focus on leveraging semantic understanding for medical concepts through PLMs. BERT (Kenton & Toutanova, 2019) and BERT* pre-train the bidirectional encoder representations from transformers on large-scale unlabeled text data to understand the context of words by training on data bidirectionally. BioBERT (Lee et al., 2020) and BioBERT* utilize biomedical text corpora for pre-training BERT, enabling improved representation of medical-specific terminology. For each visit, we first use these PLMs to encode a textual sequence formed by the names of all associated medical concepts. Then, we separately train a Multi-Layer Perceptron (MLP) based on the obtained visit embeddings for diagnosis prediction. Here, the superscript * indicates methods that incorporate patients' historical visits during training. VecoCare (Xu et al., 2023b) combines structured EHR data and clinical notes through the dual-channel retrieval mechanism to reduce heterogeneous semantic biases.

For a fair comparison, all baselines use the same EHR data without clinical notes, medications, and procedures, relying solely on the diagnostic information of patients.

## G. Experimental Results and Analyses with Top-$k$ Recall Metric

To comprehensively evaluate the performance of the proposed BoxLM, we compare it with four representative state-of-the-art baselines in Recall@$k$ ($k$=10, 20). As shown in Table 5, BoxLM consistently outperforms all baselines across both the MIMIC-III and MIMIC-IV datasets. These results are consistent with the observations reported in Section 3.2, further demonstrating the effectiveness of BoxLM.

## H. Experimental Results and Analyses on MIMIC-IV with Varying Ratio of Training Data

To investigate the impact of training data size on model performance, we conduct experiments on the MIMIC-IV dataset with varying ratios of training data, e.g., 1%, 5%, 10%, and 15% (shown in Figure 5).

In general, BoxLM significantly outperforms all baselines across all data ratios, demonstrating its ability to generalize even with limited training data. At both the visit and code levels (Figure 5(a) and Figure 5(b)), baselines show improvements as the training data ratio increases, but their performance gains are modest compared to BoxLM. For instance, BoxCare, the second-best method, improves a P@10 at the visit level from 30.60% to 35.13%, as the training data increases from 1% to 5%. However, even with 15% training data, BoxCare's performance in P@10 remains slightly below BoxLM's performance

with only 1% training data.

Additionally, the runtime analysis (Figure 5(c)) further highlights the efficiency of BoxLM. Notably, RETAIN and TRANS exhibit significantly longer training times, where RETAIN is based on the reverse time attention mechanism and TRANS relies on temporal graphical representations. In contrast, BoxLM achieves superior efficiency by leveraging box embeddings to unify structural and semantic representations in a compact and interpretable manner. This design not only reduces computational overhead but also enables faster convergence during training. Compared to BoxCare, which also utilizes box embeddings, BoxLM introduces the structure-semantic fusion mechanism and the evolve-and-memorize patient box learning mechanism. These innovations employ external semantic knowledge from PLM to further enhance efficiency. Moreover, they allow BoxLM to model temporal dynamics and hierarchical relationships with an average 33.48% lower runtime.

*Table 5.* Recall@$k$ results for diagnosis prediction (%) on the MIMIC-III and MIMIC-IV datasets with 5% training data. The best performances are highlighted in **boldface** and the second runners are underlined. *Improv.* denotes the relative improvements of our proposed BoxLM over the second runners.

| Dataset | MIMIC-III | | MIMIC-IV | |
|---|---|---|---|---|
| Metric | Recall@10 | Recall@20 | Recall@10 | Recall@20 |
| DoctorAI | 27.36 | 43.06 | 24.84 | 37.09 |
| RETAIN | 26.68 | 42.52 | 27.22 | 39.95 |
| TRANS | 27.97 | 43.82 | 23.38 | 34.84 |
| BoxCare | 28.61 | 44.63 | 29.76 | 42.33 |
| BoxLM | **34.49** | **50.71** | **34.85** | **48.65** |
| *Improv.* | 20.55% | 13.62% | 17.10% | 14.93% |

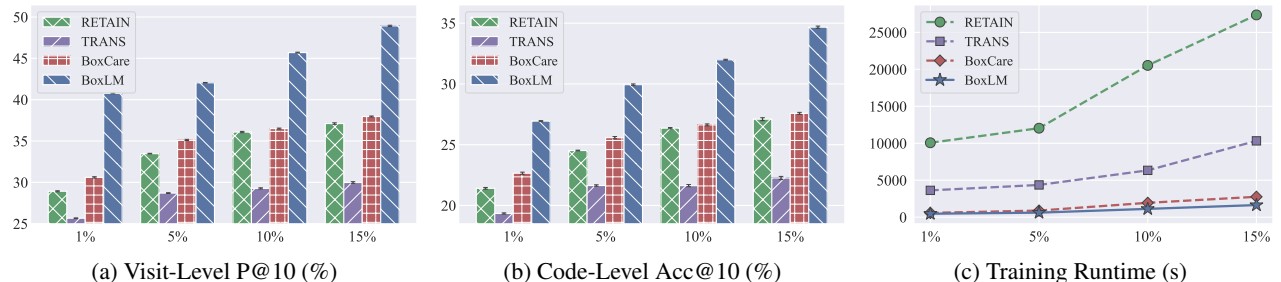

(a) Visit-Level P@10 (%)      (b) Code-Level Acc@10 (%)      (c) Training Runtime (s)

*Figure 5.* Experimental results for diagnosis prediction on the MIMIC-IV dataset with 1%, 5%, 10%, and 15% training data.

## I. Experimental Results and Analyses of Hyperparameter Studies

As shown in Table 6, we evaluate the impacts of the box embedding dimension $dim$, the Gumbel distribution scale $\beta$, and the box volume calculation. For box embedding dimension $dim$, the results show that increasing $dim$ slowly enhances model performance for both datasets. For a fair comparison with baselines, we set $dim = 16$. For the Gumbel distribution scale $\beta$, a larger $\beta$ makes the distribution closer to uniform, while a smaller $\beta$ causes its probability density function to approach a hinge function, leading the random variable to degenerate into a constant (Lin et al., 2024). In this paper, we set $\beta = 0.2$, balancing model accuracy with the distinctiveness among boxes. For the box volume calculation, we compare the soft volume calculation (Li et al., 2019) with our used Bessel volume calculation (cf., Eq. (9)). They both effectively mitigate the training difficulties that arise when disjoint boxes should overlap.

## J. Experimental Results and Analyses with Top-$k$ Consistency Metric

In our study, CCS (Clinical Classification Software) codes from the MIMIC-IV dataset can be grouped into 22 distinct body systems[3], serving as hierarchical classification labels for the predicted CCS codes. To better assess the interpretability of

---

[3]https://hcup-us.ahrq.gov/toolssoftware/ccsr/ccs_refined.jsp

BoxLM, we introduce the Consistency@$k$ metric ($k$=10, 20) (Patel et al., 2022; Xiong et al., 2022), which quantifies the alignment between model predictions and the underlying hierarchical structure. The metric can be computed as:

$$\text{Consistency@}k = \frac{1}{\min(k, |H_v|)} \sum_{i=1}^{k} \mathbb{I}(\hat{h}_i = h_i), \tag{15}$$

where $|H_v|$ is the number of ground-truth hierarchical labels associated with the CCS codes in visit $v$, and the numerator counts the number of correct hierarchical predictions in the top-$k$. We compare our proposed BoxLM with three representative state-of-the-art baselines. Table 7 demonstrates that BoxLM effectively preserves consistency on hierarchy, highlighting its strong interpretability and predictive capabilities.

*Table 6.* Hyperparameter study results (%) on the MIMIC-III and MIMIC-IV datasets with 5% training data.

| Dataset | MIMIC-III | | | | | | MIMIC-IV | | | | | |
|---|---|---|---|---|---|---|---|---|---|---|---|---|
| Metric | Visit-Level | | Code-Level | | Recall@10 | Recall@20 | Visit-Level | | Code-Level | | Recall@10 | Recall@20 |
| | P@10 | P@20 | Acc@10 | Acc@20 | | | P@10 | P@20 | Acc@10 | Acc@20 | | |
| $dim = 4$ | 40.33 | 47.48 | 29.53 | 46.05 | 31.08 | 46.80 | 36.14 | 43.53 | 26.24 | 40.17 | 29.37 | 42.86 |
| $dim = 8$ | 41.88 | 49.35 | 30.57 | 47.17 | 32.39 | 48.68 | 41.00 | 48.55 | 29.36 | 43.83 | 33.62 | 47.82 |
| $dim = 16$ | 43.88 | 51.62 | 31.74 | 48.74 | 34.49 | 50.71 | 42.04 | 49.65 | 29.94 | 44.52 | 34.85 | 48.65 |
| $dim = 32$ | **44.84** | **51.68** | **32.52** | **48.99** | **34.88** | **50.99** | **50.72** | **57.48** | **35.74** | **51.01** | **42.31** | **56.71** |
| $\beta = 0.1$ | 42.91 | 50.12 | 31.05 | 47.47 | 33.50 | 49.48 | 41.10 | 48.92 | 29.18 | 43.43 | 34.27 | 47.98 |
| $\beta = 0.2$ | 43.88 | 51.62 | 31.74 | 48.74 | 34.49 | 50.71 | **42.04** | **49.65** | **29.94** | **44.52** | **34.85** | **48.65** |
| $\beta = 0.3$ | 45.39 | 52.37 | 32.81 | 49.55 | 35.56 | 51.69 | 41.71 | 49.35 | 29.78 | 44.30 | 34.79 | 48.60 |
| $\beta = 0.4$ | 45.55 | 52.73 | 33.00 | 49.80 | **35.68** | 52.05 | 41.96 | 49.64 | 29.89 | 44.41 | 34.50 | 48.34 |
| $\beta = 0.6$ | **45.81** | **52.84** | **33.25** | **50.05** | 35.66 | **52.14** | 40.70 | 48.06 | 29.14 | 43.23 | 33.63 | 47.08 |
| Bessel Volume | 43.88 | **51.62** | 31.74 | **48.74** | **34.49** | **50.71** | 42.04 | 49.65 | 29.94 | 44.52 | 34.85 | 48.65 |
| Soft Volume | **43.91** | 51.05 | **31.79** | 48.22 | 34.31 | 50.39 | **42.23** | **49.76** | **30.10** | **44.62** | **34.93** | **48.74** |

*Table 7.* Consistency@$k$ results for diagnosis prediction (%) on the MIMIC-IV datasets with 5% training data. The best performances are highlighted in **boldface** and the second runners are underlined. *Improv.* denotes the relative improvements of our proposed BoxLM over the second runners.

| Dataset | MIMIC-IV | |
|---|---|---|
| Metric | Consistency@10 | Consistency@20 |
| RETAIN | 41.81 | 46.55 |
| TRANS | 35.13 | 38.43 |
| BoxCare | 47.36 | 52.06 |
| BoxLM | **56.53** | **62.44** |
| *Improv.* | 19.36% | 19.94% |

