# OpenReview forum: "BoxLM: Unifying Structures and Semantics of Medical Concepts for Diagnosis Prediction in Healthcare"
_ICML.cc/2025/Conference — ICML 2025 poster_

### Official Review · Reviewer_yPUD · 2025-03-12

**Overall Recommendation:** 3

**Summary:**

This paper introduces BoxLM, a novel framework for diagnosis prediction in healthcare that aims to unify the semantic understanding of medical concepts with their underlying structural relationships. This approach integrates ontology-driven hierarchies and EHR-driven visit patterns with semantic embeddings from pre-trained LMs, and it further proposes a box-aware diagnosis prediction module and volume-based similarity measurement to model the temporal dynamics of patient visits and enable accurate diagnosis prediction. Overall, this paper has a clear motivation, a relatively complete theoretical section, and provides sufficiently comprehensive experimental results.

## update after rebuttal

The authors have addressed my comments and I will maintain the original score.

**Claims And Evidence:**

BoxLM's main claim of outperforming baselines for diagnosis prediction is well-supported by MIMIC-III/IV experiments. Evidence is strong across metrics and data settings, including few-shot. Ablations and case studies validate components. However, the introduction overstates the novelty of box embeddings for hierarchical concepts, which has been proposed by BoxCare. The introduction should acknowledge BoxCare and clarify BoxLM's specific innovations. Despite this minor intro overclaim, core claims are valid. BoxLM builds on and improves prior work, showing strong results. Evidence supports the paper's conclusions.

**Essential References Not Discussed:**

No essential related works appear to be missing.

**Experimental Designs Or Analyses:**

The experimental designs are well-controlled, and the analyses are appropriate for supporting the claims of the paper. No major issues are identified in the experimental design or analyses.

**Methods And Evaluation Criteria:**

The proposed methods and evaluation criteria are appropriate and make sense. The use of MIMIC-III and MIMIC-IV is well-justified and provides a realistic benchmark. The evaluation metrics are standard metrics in the field of EHR-based diagnosis prediction and are suitable for assessing the performance of the models.

**Other Comments Or Suggestions:**

NA

**Other Strengths And Weaknesses:**

## Strength

- The paper is well-motivated, clearly articulating the research problem. The explanation of the proposed methods and formulas is presented with sufficient clarity, facilitating understanding.
- The experimental evaluation yields compelling results, demonstrating consistent and statistically significant outperformance against a comprehensive suite of baselines across two large-scale, real-world EHR datasets. This robust empirical validation strongly supports the proposed approach.
- The runtime analysis indicates that BoxLM achieves competitive computational efficiency, and even outperforms some baselines in terms of speed. This efficiency is a significant advantage for practical deployment and real-world application within clinical settings.

## Weakness

- While the paper effectively leverages box embeddings to model hierarchical medical concepts, building upon the foundational work of BoxCare, the primary innovation appears to reside in the enhanced integration of EHR information. The framing in the introduction may inadvertently overemphasize the novelty of the core box embedding concept itself, potentially leading to a perception of overclaim regarding the overall contribution relative to BoxCare.
- To improve initial reader comprehension, the paper would benefit from a more explicit and upfront articulation of the specific task being addressed and the primary research objectives immediately following the introduction. This would provide essential context and guide the reader more effectively.
- While box embeddings offer advantages, they are inherently more complex than traditional vector embeddings. The paper could benefit from further simplification or more intuitive explanations of box embedding concepts and operations, especially for readers less familiar with this technique.
- While the paper mentions setting the embedding dimension and Gumbel scale, there appears to be limited discussion on the sensitivity analysis of these hyperparameters, particularly those specific to box embeddings (e.g., dimensionality, different distance/volume metrics). Exploring the impact of these choices would strengthen the robustness of the findings.
- While paper claims interpretability, the authors should provide an example in the experimental section to visualize how different levels of concepts are represented in box embeddings, thereby enhancing the interpretability of BoxLM.

**Questions For Authors:**

See weakness.

**Relation To Broader Scientific Literature:**

This work significantly builds upon the recent advancements in representing medical concepts and their relationships using box embeddings, most notably following the approach introduced by BoxCare. It extends BoxCare by more comprehensively leveraging hierarchy from both ontologies and EHR data. Crucially, it integrates pre-trained language models by leveraging their inherent prior knowledge.

**Theoretical Claims:**

The proofs in Appendix C are correct.

---

> ### Author Rebuttal · Authors · 2025-03-31
>
> We thank the reviewer for the overall positive evaluations and many detailed suggestions. In the following, we focus on several of the main issues to provide our feedback:
>
> > **Reviewer yPUD.W1:** The difference between BoxLM and BoxCare.
>
> As described in Section 2, BoxCare only uses box embeddings to represent ontology hierarchies without fully integrating the rich semantics and hierarchical structures inherent in EHRs. In contrast, BoxLM innovatively integrates PLM with box embeddings, unifying ontology-driven and EHR-driven hierarchies with LM-based semantics for more accurate diagnosis prediction. We will clarify the difference in the introduction section.
>
> > **Reviewer yPUD.W2:** Explicitly stating the specific task and research objectives upfront after the introduction for better reader comprehension.
>
> We have provided the problem definition and an overview of BoxLM in Section 3.1. To further improve readability, we will reorganize the paper structure by moving the methodology section to Section 2, immediately following the introduction.
>
> > **Reviewer yPUD.W3:**  Box embedding concepts and operations are relatively complex.
>
> Box embeddings only introduce **one** **additional offset embedding** compared with traditional vector representation, and the time complexity for diagnosis prediction remains comparable to that of traditional vector-based approaches. Specifically, the time complexity is $\mathcal{O}(N \times C \times d)$, scaling linearly with the embedding dimension $d$, where $N$ and $C$ denote the number of patients and CCS codes, respectively.
>
> > **Reviewer yPUD.W4:** Adding hyperparameter studies.
>
> We have evaluated the impacts of the box embedding dimension $d$ (Table 1), the Gumbel distribution scale $\beta$ (Table 2), and the box volume calculation (Table 3), which will be included in the final version.
>
> For box embedding dimension $d$, the results show that increasing $d$ slowly enhances model performance for both datasets. For a fair comparison with baselines, we set $d=16$.
>
> **Table 1: Experimental results (%) for diagnosis prediction with varying box dimension $d$.**
>
> | Dataset  | MIMIC-III |        | MIMIC-IV |        |
> | -------- | --------- | ------ | -------- | ------ |
> | Metric   | P@10      | Acc@10 | P@10     | Acc@10 |
> | $d=4$  | 40.33     | 29.53  | 36.14    | 26.24  |
> | $d=8$  | 41.88     | 30.57  | 41.00    | 29.36  |
> | $d=16$ | 43.88     | 31.74  | 42.04    | 29.94  |
> | $d=32$ | 44.84     | 32.52  | 50.72    | 35.74  |
>
> For the Gumbel distribution scale $\beta$, a larger $\beta$ makes the distribution closer to uniform, while a smaller $\beta$ causes its probability density function to approach a hinge function, leading the random variable to degenerate into a constant [1]. In this paper, we set $\beta=0.2$, balancing model accuracy with the distinctiveness among boxes.
>
> **Table 2: Experimental results (%) for diagnosis prediction with varying Gumbel distribution scale $\beta$.**
>
> | Dataset     | MIMIC-III |        | MIMIC-IV |        |
> | ----------- | --------- | ------ | -------- | ------ |
> | Metric      | P@10      | Acc@10 | P@10     | Acc@10 |
> | $\beta=0.1$ | 42.91     | 31.05  | 41.10    | 29.18  |
> | $\beta=0.2$ | 43.88     | 31.74  | 42.04    | 29.94  |
> | $\beta=0.4$ | 45.55     | 33.00  | 41.71    | 29.78  |
> | $\beta=0.6$ | 45.81     | 33.25  | 40.70    | 29.14  |
>
> For the box volume calculation, we compare the soft volume calculation [2] with our used Bessel volume calculation (i.e., Eq. (9)). They both effectively mitigate the training difficulties that arise when disjoint boxes should overlap.
>
> **Table 3: Experimental results (%) for diagnosis prediction with varying box volume calculation.**
>
> | Dataset       | MIMIC-III |        | MIMIC-IV |        |
> | ------------- | --------- | ------ | -------- | ------ |
> | Metric        | P@10      | Acc@10 | P@10     | Acc@10 |
> | Soft volume   | 43.91     | 31.79  | 42.23    | 30.10  |
> | Bessel volume | 43.88     | 31.74  | 42.04    | 29.94  |
>
> **Reference**
> [1] When Box Meets Graph Neural Network in Tag-aware Recommendation, KDD, 2024.
> [2] Smoothing the geometry of probabilistic box embeddings, ICLR, 2018.
>
> > **Reviewer yPUD.W5:** Providing examples for the interpretability of the proposed method.
>
> We have provided a real case study in Section 4.3, visualizing different levels of medical concepts (e.g., diagnosis, CCS, patient). As shown in Lines 403-420, BoxLM accurately predicts the development of *coronary atherosclerosis and other heart diseases* (CCS code 101) for the patient diagnosed with *essential* *hypertension* by calculating the overlap between patient and CCS boxes, which aligns with clinical knowledge that hypertension increases the risk of atherosclerosis.

---

> > ### Comment · Reviewer_yPUD · 2025-04-03
> >
> > Thanks author's rebuttal. The authors have addressed my comments and I will maintain the original score.

---

> > > ### Author Response · Authors · 2025-04-03
> > >
> > > Dear reviewer yPUD,
> > >
> > > We thank your response and appreciation of our work and rebuttal. We will make sure to incorporate the new results and discussions into the final version.
> > >
> > > Best,
> > >
> > > Authors

---

### Official Review · Reviewer_UQNn · 2025-03-17

**Overall Recommendation:** 3

**Summary:**

In this paper, the authors focus on a critical task: diagnosis prediction. They introduce a unique approach, the BoxLM representation, to represent Electronic Health Records (EHR) and diseases. While the paper is well-written and easy to follow, it has two significant shortcomings: 1. The evaluation metric used is not comprehensive. Typically, recall is considered an essential metric in AI-assisted diagnosis, as missing a diagnosis can have serious implications. 2. There is a lack of discussion and comparison with existing approaches in diagnosis prediction.

**Claims And Evidence:**

yes

**Essential References Not Discussed:**

Moreover, the paper lacks comparison with existing works on diagnosis prediction, such as [1] and subsequent works referencing [1].

[1] Choi, Edward, et al. "Doctor AI: Predicting Clinical Events via Recurrent Neural Networks." Machine Learning for Healthcare Conference. PMLR, 2016.

**Experimental Designs Or Analyses:**

The experimental design also has some flaws. In line with previous works, the authors should include recall as an evaluation metric.

**Methods And Evaluation Criteria:**

yes

**Other Comments Or Suggestions:**

no

**Other Strengths And Weaknesses:**

no

**Questions For Authors:**

1. In Section 4.3's ablation study, the contribution of BoxLM should be evaluated. For instance, it could be compared with the classic vector representation.

2. What's the difference between CCS and diagnosis

**Relation To Broader Scientific Literature:**

no

**Theoretical Claims:**

N/A

---

> ### Author Rebuttal · Authors · 2025-03-31
>
> We thank the reviewer for the overall summary. As for the several weaknesses and questions you mentioned, our responses are listed as follows:
>
> > **Reviewer UQNn.W1:** The evaluation metric suggests including recall.
>
> Following [2-6], we adopt visit-level Precision@k (P@k) and code-level Accuracy@k (Acc@k) for the comprehensive evaluation of diagnosis prediction from coarse-grained and fine-grained perspectives, respectively (detailed in Appendix D). Additionally, we have included Recall@k (k=10, 20) in our evaluation (Table 1), further demonstrating the effectiveness of BoxLM.
>
> **Table 1: Recall@k results (%) for diagnosis prediction on both datasets with 5% training data.**
>
> | Dataset |               | MIMIC-III     |       |        |
> | ------- | ------------- | ------------- | ----- | ------ |
> | Metric  | **Recall@10** | **Recall@20** | P@10  | Acc@10 |
> | RETAIN  | 26.68         | 42.52         | 34.73 | 25.73  |
> | TRANS   | 27.97         | 43.82         | 36.64 | 26.96  |
> | BoxCare | 28.61         | 44.63         | 38.21 | 28.12  |
> | BoxLM   | 34.49         | 50.71         | 43.88 | 31.74  |
> | **Dataset** |               | **MIMIC-IV**      |       |        |
> | Metric  | **Recall@10** | **Recall@20** | P@10  | Acc@10 |
> | RETAIN  | 27.22         | 39.95         | 33.47 | 24.52  |
> | TRANS   | 23.38         | 34.84         | 28.70 | 21.64  |
> | BoxCare | 29.76         | 42.33         | 35.13 | 25.58  |
> | BoxLM   | 34.85         | 48.65         | 42.04 | 29.94  |
>
> **Reference**
> [2] Unveiling Discrete Clues: Superior Healthcare Predictions for Rare Diseases, WWW, 2025.
> [3] Stage-Aware Hierarchical Attentive Relational Network for Diagnosis Prediction, TKDE, 2024.
> [4] Predictive Modeling with Temporal Graphical Representation on Electronic Health Records, IJCAI, 2024.
> [5] MEGACare: Knowledge-guided Multi-view Hypergraph Predictive Framework for Healthcare, Information Fusion, 2023.
> [6] INPREM: An Interpretable and Trustworthy Predictive Model for Healthcare, KDD, 2020.
>
> > **Reviewer UQNn.W2:** Missing baseline Doctor AI [1].
>
> We have included Doctor AI [1] in our benchmarks for comparison. As shown in Table 2, BoxLM outperforms Doctor AI by up to 40.30% gains in Recall@10 on MIMIC-IV. Notably, both Doctor AI and RETAIN [7], developed by the same research group, use RNNs to capture the temporal dynamics of patient visits. Doctor AI slightly surpasses RETAIN with short visit sequences on MIMIC-III, while RETAIN performs better on MIMI-IV via its reverse time attention mechanism. In contrast, BoxLM integrates temporal modeling with richer representations of medical concepts from EHR data and ontological sources, leading to superior generalization. We will add this discussion to the final version.
>
> **Table 2: Experimental results (%) for diagnosis prediction on both datasets with 5% training data.**
>
> | Dataset       |           | MIMIC-III |       |        |
> | ------------- | --------- | --------- | ----- | ------ |
> | Metric        | Recall@10 | Recall@20 | P@10  | Acc@10 |
> | **Doctor AI** | 27.36     | 43.06     | 35.72 | 26.31  |
> | RETAIN        | 26.68     | 42.52     | 34.73 | 25.73  |
> | BoxLM         | 34.49     | 50.71     | 43.88 | 31.74  |
> | **Dataset**       |           | **MIMIC-IV**  |       |        |
> | Metric        | Recall@10 | Recall@20 | P@10  | Acc@10 |
> | **Doctor AI** | 24.84     | 37.09     | 30.65 | 22.69  |
> | RETAIN        | 27.22     | 39.95     | 33.47 | 24.52  |
> | BoxLM         | 34.85     | 48.65     | 42.04 | 29.94  |
>
> **Reference**
> [7] RETAIN: An Interpretable Predictive Model for Healthcare using Reverse Time Attention Mechanism, NeurIPS, 2016.
>
> > **Reviewer UQNn.Q1:** In Section 4.3's ablation studies, the contribution of BoxLM should be evaluated, including the comparison with the classic vector representation.
>
> Our ablation studies in Section 4.3 have included the comparison with classic vector representations (i.e., the "Base" model in Table 3). Specifically, the "Base" model uses BioBERT embeddings without structure modeling, which represents visits as single points. By comparison, "+ Onto" and "+ EHR" introduce box embeddings for CCS codes, diagnoses, and visits with our ontology-based and EHR-based hierarchy modeling, achieving significant improvement by up to 25.33% gains in P@10 on MIMIC-IV. These experimental results provide a comprehensive understanding of the importance and reliability of our designed techniques.
>
> > **Reviewer UQNn.Q2:** What's the difference between CCS and diagnosis?
>
> Sorry for the confusion. CCS is a classification system that groups diagnosis codes into clinically meaningful categories for analytical purposes, whereas a diagnosis refers to the identification of a specific disease or condition in a patient's visit. We will add this explanation in the final version.

---

> > ### Comment · Reviewer_UQNn · 2025-04-07
> >
> > The authors have addressed my concerns in their rebuttal, I increase the score.

---

> > > ### Author Response · Authors · 2025-04-07
> > >
> > > Dear reviewer UQNn,
> > >
> > > We really thanks for your time in the review and discussion. We will make sure to properly incorporate the new results and discussions into the final revision.
> > >
> > > Best,
> > >
> > > Authors

---

### Official Review · Reviewer_skUY · 2025-03-25

**Overall Recommendation:** 3

**Summary:**

The paper proposes BoxLM, a framework unifying structural (ontology and EHR hierarchies) and semantic (language model embeddings) representations of medical concepts using box embeddings. Key contributions include a structure-semantic fusion mechanism, an evolve-and-memorize patient box learning module for temporal dynamics, and volume-based similarity for diagnosis prediction. Experiments on MIMIC-III/IV datasets demonstrate state-of-the-art performance, particularly in few-shot scenarios. The authors emphasize interpretability through geometric overlap analysis and case studies.

## update after rebuttal

Authors addressed most of my concerns, I have raised my Overall Recommendation score from 2 to 3.

**Claims And Evidence:**

Mostly yes. Interpretability claims rely on a single case study (Figure 4); quantitative metrics for interpretability (e.g., user studies or consistency scores) are absent.

**Essential References Not Discussed:**

N/A

**Experimental Designs Or Analyses:**

- Validity: 5-fold cross-validation and Adam optimizer are standard. Few-shot experiments (1%–15% training data) robustly demonstrate generalizability.
- Reproducibility: Runtime comparisons (Figure 3c) lack hardware details. Code availability is promised but not provided.

**Methods And Evaluation Criteria:**

Generally make sense.
- Box embeddings effectively model hierarchical inclusion and EHR-driven visit patterns. The fusion of ontology/EHR structures and evolve-and-memorize mechanism are novel and appropriate.

**Other Comments Or Suggestions:**

- Typos: "Exampled" → "Example" (Figure 4 caption).
- Section 3.3.1 needs a diagram for patient box evolution.
- Include ablation on box dimensionality (see Questions).

**Other Strengths And Weaknesses:**

My concerns are mainly the interpretability, and your motivation for designing the method. It might not be proper to say`Luckily, we find that box embeddings offer a promising solution by representing entities as highdimensional hyperrectangle.` as it's not straightforward to encode highdimensional embeddings to hyperrectangle from my side.

**Questions For Authors:**

- How does BoxLM handle missing or incomplete hierarchical links in EHRs (e.g., unrecorded parent-child relationships)?
- What is the impact of box dimensionality (dim) on performance and computational cost?

**Relation To Broader Scientific Literature:**

Properly credits prior work (e.g., BoxCare, BioBERT) but might overlook recent geometric approaches in other fields (not limited to EHR / healthcare, I'm not sure).

**Theoretical Claims:**

N/A

---

> ### Author Rebuttal · Authors · 2025-03-31
>
> We thank the reviewer for the constructive comments on our paper. In response to the shortcomings mentioned, we provide the following answers to several questions:
>
> > **Reviewer skUY.W1:** Adding quantitative metrics for interpretability.
>
> In our study, CCS (Clinical Classification Software) codes from the MIMIC-IV dataset can be grouped into 22 distinct body systems, serving as hierarchical classification labels for the predicted CCS codes. To assess the interpretability of BoxLM, we have introduced the **Consistency@k** metric (k=10, 20) [1, 2], which quantifies the alignment between model predictions and the underlying hierarchical structure.
>
> The metric can be computed as: $Consistency@k=\frac{1}{\min(k,|H_v|)}\sum_{i=1}^k\mathbb{I}(\hat{h}_i=h_i)$, where $|H_v|$ is the number of ground-truth hierarchical labels associated with the CCS codes in visit $v$, and the numerator counts the number of correct hierarchical predictions in the top-k.
>
> The results (Table 1) demonstrate that BoxLM preserves consistency on hierarchy.
>
> **Table 1: Consistency@k results (%) for diagnosis prediction.**
>
> | Dataset |                    | MIMIC-IV           |       |        |
> | ------- | ------------------ | ------------------ | ----- | ------ |
> | Metric  | **Consistency@10** | **Consistency@20** | P@10  | Acc@10 |
> | RETAIN  | 41.81              | 46.55              | 33.47 | 24.52  |
> | TRANS   | 35.13              | 38.43              | 28.70 | 21.64  |
> | BoxCare | 47.36              | 52.06              | 35.13 | 25.58  |
> | BoxLM   | 56.53              | 62.44              | 42.04 | 29.94  |
>
> **Reference**
> [1] Hyperbolic Embedding Inference for Structured Multi-Label Prediction, NeurIPS, 2022.
> [2] Modeling Label Space Interactions in Multi-Label Classification Using Box Embeddings, ICLR, 2021.
>
> > **Reviewer skUY.W2:** Lacking hardware details and code availability.
>
> We have provided the code link in the bottom left of Page 6 for reproducibility. Additionally, we have added the following hardware details: All experiments were conducted on two NVIDIA GTX 3090 Ti GPUs.
>
> > **Reviewer skUY.W3:** Adding related works on recent geometric approaches.
>
> We have included relevant work on box embeddings from other fields (see Appendix A). Additionally, we will add the discussion and proper citations for [3, 4] in our related work, covering polygon and hypercube embeddings in areas such as knowledge graphs and recommendation systems.
>
> **Reference**
> [3] ExpressivE: A Spatio-Functional Embedding for Knowledge Graph Completion, ICLR, 2023.
> [4] Thinking inside The Box: Learning Hypercube Representations for Group Recommendation, SIGIR, 2022.
>
> > **Reviewer skUY.W4:** The saying for introducing box embedding might not be proper, as it's not straightforward to encode highdimensional embeddings to hyperrectangle.
>
> We understand that the sentence "... representing entities as high-dimensional hyperrectangles" (Lines 67-68) may not have clearly conveyed our design. Specifically, we propose using box embeddings to capture complex relationships (e.g., inclusion and intersection) in medical data. This representation aligns well with the hierarchical structure of medical ontologies and EHR-driven visit patterns. We will revise this sentence in the final version.
>
> > **Reviewer skUY.W5:** Some typos exist.
>
> Thanks for pointing this out. We will fix it in the final version.
>
> > **Reviewer skUY.W6:** Section 3.3.1 needs a diagram for patient box evolution.
>
> We have included a diagram for the evolve-and-memorize patient box learning mechanism in the upper right of Figure 2. For clarity, we will explicitly reference it in Section 3.3.1.
>
> > **Reviewer skUY.Q1:** How does BoxLM handle missing or incomplete hierarchical links in EHRs?
>
> BoxLM uses box embeddings to model medical concepts from ontology and EHR data. The geometric properties of these embeddings allow the model to infer implicit relational structures, such as transitivity and entailment. This capability, demonstrated in tasks like taxonomy expansion [5] and knowledge base completion [6], enables BoxLM to recover missing or incomplete hierarchical links.
>
> **Reference**
> [5] A Single Vector Is Not Enough: Taxonomy Expansion via Box Embeddings, WWW, 2023.
> [6] BoxE: A Box Embedding Model for Knowledge Base Completion, NeurIPS, 2020.
>
> > **Reviewer skUY.Q2:** What is the impact of box dimensionality on performance and computational cost?
>
> We have added experiments to analyze the impact of box embedding dimension $d$ (please refer to Table 1 in **Reviewer yPUD.W4**), which will be included in the final version. The results show that increasing $d$ slowly enhances model performance for both datasets. Notably, the time complexity of diagnosis prediction scales as $\mathcal{O}(N \times C \times d)$ ($N$ and $C$ denote the number of patients and CCS codes), which grows linearly with $d$ and is similar to traditional vector-based approaches. For a fair comparison with baselines, we set $d=16$.

---

> > ### Comment · Reviewer_skUY · 2025-04-03
> >
> > Thanks for your response. I have raised my scores accordingly.

---

> > > ### Author Response · Authors · 2025-04-03
> > >
> > > Dear reviewer skUY,
> > >
> > > Thank you for your feedback. We are glad our response addressed your concerns. It seems the scores haven't been updated in the system yet. We'd be grateful if you could kindly double-check when convenient.
> > >
> > > Best,
> > >
> > > Authors

---

### Decision · Program_Chairs · 2025-05-01

**Decision:**

Accept (poster)

**Comment:**

The three reviewers are uniformly positive about the paper, rating it as 'weak accept' as a final recomendation.
The AC agrees with the reviewers and recommend 'Accept'.